# A Qualitative Study of the Health Perceptions in the Venezuelan Immigrant Population in Medellín (Colombia) and Its Conditioning Factors

**DOI:** 10.3390/ijerph18083897

**Published:** 2021-04-08

**Authors:** Andrés M. Murillo-Pedrozo, Eliana Martínez-Herrera, Elena Ronda-Pérez, Andrés A. Agudelo-Suárez

**Affiliations:** 1Faculty of Dentistry, University of Antioquia, Medellín 050010, Colombia; andresmauricio.murillo@gmail.com; 2Research Group of Epidemiology, National School of Public Health “Héctor Abad Gómez”, University of Antioquia, Medellín 050010, Colombia; eliana.martinez@udea.edu.co; 3Research Group on Health Inequalities, Environment, Employment Conditions (GREDS-EMCONET), Department of Political and Social Sciences, Universitat Pompeu Fabra, 08005 Barcelona, Spain; 4Public Health Research Group, University of Alicante, 03690 Alicante, Spain; elena.ronda@ua.es; 5CIBERESP, 28029 Madrid, Spain

**Keywords:** emigration and immigration, oral health, mental health, health services accessibility, qualitative research

## Abstract

This study explored the general and oral health perceptions in the Venezuelan immigrant population in Medellín (Colombia) and its conditioning factors. A qualitative study involving Venezuelan immigrants ≥18 years with a minimum stay of six months in Colombia was conducted. Dentists, dental students, and other health professionals also participated. Semi-structured interviews (*n* = 17), focus groups (*n* = 2), and key informants’ interviews (*n* = 4) were utilized. The interviews and focus groups were recorded and transcribed for later narrative content analysis. A high degree of vulnerability of participants was found due to the precarious living conditions from the premigratory moment and the lack of job placement possibilities at the time of settling in Colombia, where the migratory status played a fundamental role. Among the perceived needs, the mitigation of noncommunicable diseases stood out. Poor mental health symptoms (depression and anxiety) were perceived, and oral health was not a priority. Barriers to accessing health and dental care were found. The migrant condition was found to be a determinant that affected physical, mental, and oral health and the provision of health care. This situation is of interest to the construction of public health policies that guarantee access to fundamental rights.

## 1. Introduction

Migration as a social phenomenon represents a challenge for public health in demographic, political, and economic terms for the sending and receiving countries of the migrant population. Current data provided by the International Organization for Migration and other organizations estimate that the number of international migrants at midyear 2020 was about 280.6 million persons, which was equivalent to 3.6% of the total world population [1]. It is important to mention that within the international migratory movements, there is a classic pattern of mobility from low- and middle-income countries to high-income countries (south–north migration) and another pattern characterized by movements between countries with similar income levels (south–south migration).

One of the most important migratory phenomena in recent years is the massive departure of the Venezuelan population as a result of changes in the political and social order [2]. According to available data, an estimated 5.1 million Venezuelan migrants, refugees, and asylum seekers have been reported by host governments [3]. Migración Colombia—Ministerio de Relaciones Exteriores (Colombia Migration—Ministry of Foreign Affairs) reported that over 1.7 million Venezuelans are in Colombia, which is a third of its global migrant population [4]. Regarding specific research about the health situation and related factors in this population in Colombia, some studies with varying methodological approaches have been carried out, such as case studies and descriptive studies, to investigate the health needs of the Venezuelan population in particular contexts and the precarious employment and its impact on the health of the working population [5,6,7]. However, it seems necessary to provide further scientific evidence regarding many aspects that have not been answered, taking into account social determinants and other particularities in this population.

Generally, the relationship between migration and health has had a great impact on the scientific agenda in recent decades. This is evidenced in studies on indicators of general health, occupational health [8,9], mental health [10,11], and sexual and reproductive health [12], as well as the use and utilization of health services [13] and access barriers [14]. Specifically, in oral health, studies have analyzed the indicators related to, among others, dental caries [15,16]; periodontal disease [16]; oral-health-related quality of life [17]; inequalities in the use of and access barriers to dental care [18,19]; the impact of acculturation in oral health [20]; beliefs, knowledge, and practices in oral health [21]. Research has been focused on describing the health conditions of specific migrant groups [22,23] or making comparisons between migrants and their native counterparts [11,13].

Currently, this global scientific panorama has changed substantially due to the coronavirus disease 2019 (COVID-19) pandemic (caused by severe acute respiratory syndrome coronavirus 2 (SARS-CoV-2)) [24]. In this context, there have been profound transformations in the economic, labor, and health fields, and this may be more evident in groups with high social vulnerability, which in this case, is the Venezuelan immigrant population that has remained in Colombia [25].

Qualitative research has been implemented with greater scientific agenda growth in the fields of dentistry and public health [26,27]. This kind of research allows for the recognition of opinions and perceptions regarding oral health and its determinants from a social perspective, enabling new approaches toward understanding various health phenomena. Similarly, it seems important to recognize, from an integral aspect, the relationships that exist between oral health and general health, making a social construction of the problem. Nevertheless, qualitative research focused on the study of migration and oral health in the Latin American context is scarce [28]. Accordingly, this study aimed to explore the perceptions of the state of oral and general health in the Venezuelan immigrant population in Medellín (Colombia) and its conditioning factors.

## 2. Materials and Methods

### 2.1. Approach and Design

For the development of this study, an open and flexible design was proposed, which generated an approximation of the participants’ perceptions. Hence, a qualitative and exploratory approach was employed, which provided a deep understanding of social systems, as is the case with the ethnographic perspective [29]. This approach has a contextual and holistic nature and is descriptive and reflective. According to the participants’ characteristics, their homogeneity in social and demographic terms, and the particularities of the phenomenon to be investigated, a focused ethnography approach was adopted [30].

### 2.2. Participants

Information was obtained from the discourse analysis of the experiences of three main sources:Venezuelan immigrants: The study included 17 Venezuelan immigrants of legal age (≥18 years old); men and women with a minimum stay of six months in Colombia and a minimum labor experience of one month were included. Only Venezuelan nationals were considered. In this participation criterion, those who did not have a Colombian citizenship identification card and who had less opportunity for access to various social and health services were taken into account. Those who were Colombian nationals or children of Colombians did not participate. There were no restrictions on immigration status (regular or irregular) for participating. For the selection of participants, theoretical and/or intentional sampling strategies, including snowball sampling and referrals, were used, and patients included those who consulted the emergency service of the Faculty of Dentistry, University of Antioquia, and other institutions that were in contact with the immigrant population. In each case, the voluntary nature of participation was regarded. The final number of participants was determined via saturation, that is, verifying that no new data for the objectives of the study emerged [31]. The sociodemographic characteristics of the Venezuelan participants are summarized in Table 1.

Key informants: Four health professionals (one medical doctor, one psychologist, and two dentists) from two institutions that were consulted by the immigrant population participated, offering their experiences when assisting the Venezuelan population in Medellín.Dental staff: Dentists (*n* = 6) and dental students (*n* = 5) from the University of Antioquia participated in two focus groups (FGs) to identify the social and contextual determinants of the immigrant population from the perspective of the people who served this population type in public and private health institutions. These FGs were enough to collect information from the dental staff’s perspectives according to the research objectives.

### 2.3. Techniques and Instruments of Information Gathering

The fieldwork took place between January and December 2020. Semi-structured interviews were conducted with the immigrant population and key informants. The interviews were based on previously established axes of analysis series (Figure 1). Open dialogue with the interviewees facilitated the emergence of multiple categories that enriched the data. Sociodemographic information about the Venezuelan participants was gathered for categorization (Table 1). In addition, two FGs were created. The research team produced a guide for use in the FGs and interviews that indicated a series of topics to be discussed among the participants. This list included the analysis of the oral health situation and its relationship with general health in the Venezuelan population in Colombia from the dental staff’s perspective, in addition to the identification of access barriers and the role of the health system as a social determinant of health.

The interviews, which lasted an average of 45 min, and the FG discussions, lasting an average of 60 min, were conducted by the first author (A.M.M.-P.). This work was conducted to fulfill the degree requirements for the Master of Dentistry Sciences program of the University of Antioquia (Medellín, Colombia). All the fieldwork was originally supervised by the last author (A.A.A.-S.), who has a Ph.D. in public health with expertise in qualitative methods and migration studies. The other authors (E.M.-H. and E.R.-P.) verified the information presented by other members of the research team.

### 2.4. Data Analysis

First, to identify the text fragments and meanings, an initial reading of the data was done. The transcribed data were imported into the qualitative analysis software ATLAS.Ti 8.4 (Scientific Software Development GmbH, Berlín, Germany), and the analysis of the codes and categories was developed via a narrative analysis. In the beginning, 55 codes relating to the objective of the study were established based on the interviews and FG discussions. This generated five final mixed categories, concept maps, and causal networks of meaning. The extracts of the discourses in the text and tables are presented using labels indicating the data sources (I—interview, KI—key informant, and FG—focus group). The number and sex of each quoted participant are also provided.

### 2.5. Ethics Approval and Consent to Participate

All procedures performed in this study were in accordance with the ethical standards for human research. This study was reviewed and approved by the Committee of Bioethics of Research at the Faculty of Dentistry of the University of Antioquia, Colombia, according to Act 14 of 2019. In accordance with Colombian legislation and in light of the methodology used, this study was considered non-risky research. Participation was voluntary and written informed consent was provided by each participant, guaranteeing the confidentiality of and respect for the information each participant provided. Lastly, this manuscript was written according to the Standards for Reporting Qualitative Research [32] and the Consolidated Criteria for Reporting Qualitative Research [33].

## 3. Results

The most important aspects that are related to the general and oral health of the migrant population are described below according to the proposed categories and those that emerged from the interviews with the study participants. Figure 2 shows a general conceptual map of the categories and codes that emerged from the analysis for the general topic: perceptions of the oral and general health status of the Venezuelan migrant population in Medellín. In addition, a crossover category of “migratory status (concerning the residence/work permission in Colombia: regular/irregular)” was established.

### 3.1. Migratory Process

When we inquired about the factors that influenced their decision to migrate, most of the participants stated that their main motivation was the search for a better future by getting a job to improve their living conditions and that of their families (Table 2: 1a,b). The structural crisis that Venezuela was going through was the main trigger for the exodus of Venezuelans, making evident the discontent of migrants with the current government (Table 2: 1c).

The Venezuelan migrant population arriving in Colombia exhibited the characteristics of a lower-middle-class population that did not have the resources to migrate by air. They generally migrated by bus, trails, and even on foot (Table 2: 1d,e).

For the choice of the destination country, the migrants took into account the proximity of the countries to them and the cost of transportation, with Latin American countries being the easiest for most Venezuelans to migrate to, especially Colombia, as it was the closest. Additionally, those who arrived in the city Medellín did so mainly because they had family members or friends who had already settled in the city that recommended that they should emigrate to it, reflecting the importance of social support networks in migration processes (Table 2: 1f,g).

### 3.2. Health

The interviewees gave different definitions of what they considered the concept of health to be. Most of them linked health to the absence of symptoms, while others related it to physical and mental well-being, quality of life, and the ability to carry out daily activities without impediments (Table 2: 2a,b).

The findings regarding the perception of the current state of the health of the migrants were diverse. While a large part of the population, generally the young adult population, reported a good state of health, the slightly more adult population presented with some chronic conditions (Table 2: 2c,d). This was complemented by inquiring about the perception of the evolution of the participants’ state of health. In many cases, there seemed to be an improvement in their state of health after their settling in Colombia, as they could access health care or medicines for chronic diseases and a better diet (Table 2: 2e).

Throughout the interviews, many of the participants agreed that oral health had not been considered a priority despite having such needs even from the beginning of their migratory process. However, once they were settled in Colombia, these oral conditions that were in the background became more acute. These difficulties in accessing oral health services in Venezuela added to the need to migrate and caused the Venezuelans to abandon their dental treatments or leave them unfinished. Colombian dentists and dental students participating in the study identified some aspects related to the emergency care associated with exacerbated processes as the main reasons for consultation of the Venezuelan population, among which, deep caries with pulpitis, root remnants, or associated dental abscess. These dental emergency issues have been the main gateway for seeking oral health care, mainly in the public sector (Table 2: 2f–h).

It is common for migrants to go through migratory grief, which affects their mental health. Even the few who had the opportunity to migrate found it necessary to seek psychological support. Cultural adaptation is the main reason migrants seek consultation, according to a foundation that helps Venezuelan migrants, and the change in their socioeconomic status was identified as the most difficult change when assimilating into their new lifestyle (Table 2: 2i–k).

### 3.3. Barriers and Facilitators of Access to Health Services

The main access barrier mentioned by the interviewees was related to their residence and documentation status. Both the migrants and the health professionals agreed that the state of irregularity was the main difficulty (Table 2: 3a,b). Not being linked to the social security system leaves migrants with private services as the only health care option they can access, with treatment costs being the main limitation, especially in dental care (Table 2: 3c).

The interview results showed that the positions occupied by the migrants were generally related to low-skilled occupations in the service sector, with informality and the lack of labor links being a constant. Thus, the migrant workers tended to avoid requesting permission to be absent from their jobs due to sickness or to attend medical consultations, as those who were employed were afraid of losing their jobs, while those who worked independently needed to work even when they were sick to get their daily income (Table 2: 3d–f).

To obtain housing, immigration status was an important influence since, as described by the participants, not having documents made it difficult to enter into lease contracts. The interviews revealed that when a migrant managed to find housing, it was common to share it with several of their family members or friends (Table 2: 3g,h).

Concerning the affiliation to the General Social Security System, the common denominator was not having health insurance. As for housing and employment, irregularity was a common feature of membership. In many cases, when a family member managed to link up and make contributions, he or she could not enroll their family because they did not have the correct documents, such as the Special Permit of Permanence (Permiso especial de permanencia in Spanish) (Table 2: 3i,j).

### 3.4. Support Networks

Generally speaking, the interviewees had good support networks that facilitated their settling and permanence in Colombia. In the first instance, the networks of family and friends who were of Venezuelan origin and neighbors in the migrants’ place of residence stood out. Furthermore, other key actors included nongovernmental organizations and churches (Table 2: 4a–c).

### 3.5. Expectations

The interviewees highlighted their different plans, which could be divided into short-, medium-, and long-term plans. In the short term, the most frequent expectation of the migrants was associated with satisfying their basic needs by getting a job or finding a home with better conditions (Table 2: 5a,b). Regarding the medium term, the interviewees planned to achieve economic stability and define their regular situation in the country to obtain a better job and, in some cases, standardize their qualifications and practice their profession in Colombia (Table 2: 5c). Regarding the long term, the expectations of the migrants were diverse. While the majority of the migrants had the conviction to settle in Colombia, others had the intention of returning to Venezuela, but they found it difficult given that the conditions in their country had not improved. According to their perception, they doubt that the condition will change for a long time (Table 2: 5d,e).

## 4. Discussion

This study explored the perceptions of the state of general health and particularly oral health of the Venezuelan migrant population and the conditioning factors. According to the findings of previous studies [7,34], migrants leave their country mainly due to a lack of work, political-ideological problems, insecurity and violence, wars, ethnic-religious persecution, and socioeconomic problems. This is consistent with the results of this study, as the structural problems of Venezuela are directly associated with the political, social, and economic instability that the country has been experiencing in recent years [7].

Generally, migrants desire to move to countries in the north or countries with a higher degree of economic development [35]. However, in the case of the Venezuelan groups, migrants tended to be lower-middle-class people who sought a destination country with geographic proximity, which is consistent with findings reported by oral health studies conducted in South Asia [22,36], allowing them to reduce travel costs and, in some cases, avoid border checkpoints. This population, as has been reported [7,8,9], tends to be in low-skilled jobs without affiliation to the social security system, which is mostly, according to our results, the state of irregularity in which most Venezuelans found themselves.

Regarding health, its concept is related to physical and mental well-being and the absence of symptoms. Likewise, the self-perception of health status in the migrant population in some studies has been reported as good or very good [7,37,38], including a study carried out among Venezuelans living in the city of Barranquilla (Colombia) [6]. In our findings, the migrants perceived themselves to be in good general health in most of the interviews. Research conducted in south–north migration contexts explained the favorability of some health indicators for recent migrants as being due to “the healthy migrant effect” [38], and this situation is attributed to a selective migratory process of healthy groups into the host countries. This is in contrast to our findings since the perception of the study participants (Venezuelans and health professionals) was that the migrants tended to settle with a burden of disease that came from the premigration process and their health status began to improve over time when they adapted to the new country. In cases where a healthy immigrant effect could occur, the period was usually short, largely due to the lack of experience in migration issues in the Colombian case and its incipient capacity for social response to the migration.

In the case of disparities and inequalities for oral health indicators in south–south contexts [28], worse oral health status has been reported among migrants in different studies [16], which is reflected in a higher frequency of dental caries compared to the autochthonous population [39], wherein access to dental care is faced with different barriers, with the main one being the economic factor. In Venezuelan migration, the main barrier to any type of health care is immigration status, and undocumented migrants are the most affected. Additionally, the lack of interest in health care plans for migrants who do not see oral health as a priority and the need to satisfy their other needs, such as housing and food, means that the oral health of migrants can suffer and it is only taken into account in cases of exacerbation [37].

Mental health is crucial in the study of migrant health since a type of chronic stress associated with migratory grief has been reported, known as Ulysses syndrome [40]. However, our findings revealed that migratory grief processes were common among the migrants; thus, there seemed to be no cases of chronic stress that would have allowed us to relate to this syndrome. This pathology is largely related to feelings of nostalgia for the family. However, the Venezuelan migration was characterized by being family migration, and having relative members within the support networks was a protective factor. In contrast, those who migrate without support networks tend to have greater difficulties in adapting to the destination country [41]. Although there was no evidence of marked chronic stress among the research participants, it is common to find different griefs and acculturation processes that affect the state of mental health. These acculturation processes have been found to be related to greater impairment of mental health, with manifestations of anxiety and depressive disorders, obesity, smoking, cardiovascular risk, and a sedentary lifestyle [42,43]. Conversely, regarding oral health, it has been shown that individuals with a high level of acculturation have fewer decayed teeth and periodontal disease and are more inclined toward the use of dental services [20]. This is an indication that in the case of Venezuelan migration in Colombia, there may be a greater impact on mental health than physical health.

An important risk factor for the health of migrants is the different barriers to accessing health services (primary health care, specialized services, and oral health care). Our results showed that immigration status was the main determining factor when trying to access health services, followed by the economic factor. The informality and lack of health insurance among the migrant population stood out, adding to the problems inherent to the Colombian health system. This aspect was mentioned by Abadía et al. [44] and called “bureaucratic itineraries,” referring to all those routes and procedures that a patient is faced with in search of medical attention to finally receiving the refusal to be provided the right to health care. Other authors, such as Tanahashi [45], treat the concept of “effective coverage” as the proportion of the population that receives effective care and proposes an access model with an interaction between the provision of the service and its availability with the population. It is important to recognize the phases that pose barriers to access (availability, accessibility, acceptability, or contact with the service) in the search for effective coverage in the case of vulnerable social groups [46]. Although the Colombian health system has adequate resources and services, adequate access to health care has not been achieved. Only those migrants with the longest stay in Colombia, better support networks, dual nationality, an employment contract, or affiliation to the General Social Security System and, consequently, better self-perception of the state of oral and general health have managed to access health services. This demonstrates the conditions of inequity concerning the most vulnerable migrants, which is known in the literature as the “inverse care law” [47], where those with the greatest health needs are those who have the least access to health services [48]. Finally, another important finding in our study in relation to access barriers is the high sickness presenteeism since, as has been reported in other studies, migrants avoid health consultations in order to not be absent from work; they go to work even when sick to preserve their jobs or not decrease their income [49]. Presenteeism is a public health concern and is associated with a loss of productivity, future sickness absenteeism, and other health affectations [49]. Further research focused on this aspect should include presenteeism by studying employment and working conditions and their impact on health in the Venezuelan population in Colombia [7].

Although exploring the perceived reality of a group of Venezuelans who migrated to Medellín is vital qualitative research, regarding the relationships between oral, general, and mental health and the associated conditioning factors, the strengths, limitations, and the scope of the study must be taken into account. The use of different information sources allowed for an accurate triangulation of the findings. The fieldwork was carried out throughout 2020, which allowed us to conduct the interviews before, during, and after the strict quarantine decreed by the National Government of Colombia in the framework of the COVID-19 pandemic. We obtained various perceptions and framed them within the prevailing social, economic, political, and epidemiological contexts. According to the scientific literature, there is a call for attention regarding the important burden of noncommunicable diseases during the COVID-19 pandemic, such as oral health [50]. As it is an exploratory study, the analysis of the categories involved a descriptive scope of a specific population; hence, the findings of this study cannot be extrapolated to the entire migrant community. Furthermore, it must be thought that the social reality may be different in other regions; it is not the same in Medellín as in other regions in Colombia. It was not an objective of the study to explore the gender category in-depth, but it should be considered of interest for future studies given the indications of our results. Moreover, further studies should include Venezuelans of dual nationality who may be vulnerable.

## 5. Conclusions

Migration is a multicausal phenomenon that is triggered by the search for better living conditions. However, Venezuelan migration has a particularity: migrants recognize the structural problems of their country and largely attribute the instability to the current government. This population consisted mostly of young people in their productive age who were in search of better working conditions. It is common to observe a migration of the family group in which the migrants are part of different age groups. They have diverse profiles of disease, where diverse health needs are presented. The lack of oral health care stood out as an area of special interest, as it was not considered a priority, either by the migrants or by the health system, despite being a frequently reported health need.

Research on migration and health requires intersectoral and joint work between academia, researchers, and decision-makers to strengthen actions in public health. The findings from this kind of research will help to provide an approach to the diagnosis of the health situation and characterization of migrants for the development of inclusive public health policies. With the results of the present study, new research perspectives are provided and opened, which include different methodological designs and the combination of quantitative and qualitative techniques to understand the social and contextual elements that affect health–disease processes.

## Figures and Tables

**Figure 1 ijerph-18-03897-f001:**
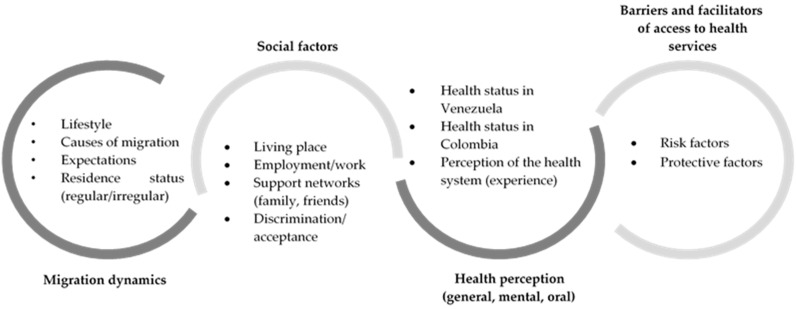
List of topics used for the fieldwork in the study.

**Figure 2 ijerph-18-03897-f002:**
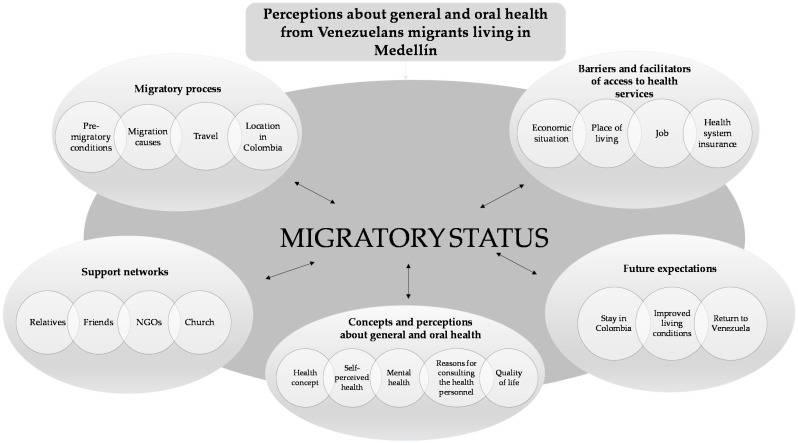
Final design for the categories based on the qualitative findings. NGOs: nongovernmental organizations.

**Table 1 ijerph-18-03897-t001:** Sociodemographic characteristics of the Venezuelan participants in Medellín, Colombia (*n* = 17).

Characteristics	*n* (%)
Sex	
Male	8 (47)
Female	9 (53)
Zone	
Urban (cities and municipalities)	14 (82)
Rural (sidewalks, villages, little towns)	3 (18)
Age (years), mean (range)	37 (19–56)
Time of residence in Colombia (months), mean (range)	24 (9–60)
Medical antecedents in Venezuela	
Yes	3 (18)
No	14 (82)
Employment situation	
Yes	7 (41)
No	10 (59)
Presence of work contract	
Yes	3 (18)
No	14 (82)
Health insurance	
Yes	6 (35)
No	11 (65)
Marital status	
Single	6 (35)
Married/cohabitated	10 (59)
Widow	2 (12)
Education	
Primary	1 (6)
Incomplete secondary	1 (6)
Complete secondary	5 (29.4)
Technical studies	5 (29.4)
University	5 (29.4)

**Table 2 ijerph-18-03897-t002:** Verbatim interview extracts from participants’ discourses. Medellín, Colombia (2020) *.

Categories	Keywords	Verbatim Extracts form Participants’ Discourses
(1) Migratory process	Motivations	(a) “[I am] seeking a better future for my family and my babies.” (I-12, female, 23 years old)
		(b) “Because in Venezuela there was no food, there was no medicine, there was nothing, looking for a better one here.” (I-6, female, 46 years old)
		(c) “Because of the situation in the country, because of the circumstances that all Venezuelans are going through, because of this government, a dictatorship.” (I-3, male, 29 years old)
	The trip	(d) “In a chirrinchera [typical Venezuelan word] … a chirrinchera is a truck that has rails on the sides, and there, one comes standing. For all that is La Guajira [Colombian region], grabbing the trails of Maicao [Colombian municipality].” (I-9, female, 56 years old)
		(e) “Along the trails and that day, a family member drowned because it was in the rain, and a family member drowned when crossing the river. We passed … Well, that was horrible.” (I-6, female, 46 years old)
	Destination choice	(f) “It is the closest place we have. Even though Brazil is also close to Venezuela, the closest is Colombia.” (I-8, female, 37 years old)
		(g) “Here in Medellín [Colombia], we had an acquaintance. He was a comadre [relative] of my wife, who was here in Medellín. She received us for three days.” (I-7, male, 33 years old)
(2) Health	Health concept	(a) “For me, health is the physical and mental state that people have and that must be guaranteed and supported by a system.” (I-4, male, 35 years old)
		(b) “To have a better quality of life, being in health, because if one does not have good health, how can one have a better quality of life to work, to do other things, to take care of even the family.” (I-14, female, 44 years old)
	Perception of current health status	(c) “Well, thank God. Since I got here, I have never had a difficulty such as telling you that I have been in bed. Thank God, no.” (I-16, male, 34 years old)
		(d) “Today, I feel a little bad, from the colon, and I don’t sleep. Sometimes I don’t sleep because sometimes it won’t let me sleep [due to] a lot of pain.” (I-2, male, 45 years old)
		(e) “Right now, it has improved because here, at least you can get the right medicine for hypertension both for me and my dad.” (I-13, male, 45 years old)
	Oral health status	(f) “I came from Venezuela. I had work done on my tooth [molar], but only halfway—that is, I could not finish it.” (I-11, male, 40 years old)
		(g) “I consider that much of this burden of diseases came from Venezuela, which obviously worsened. Yes, [due to] cavities not attended here, that come from there and are not treated here [and] the difficult access here, they became worse. That’s the reason for the emergency care for them.” (KI, dentist, female, public health professional, public sector)
		(h) “Main reasons for [dental] consultation (...) they [Venezuelans] always come for many root tips of teeth, many dental caries, many pulpitis …” (FG-1, dentist, public and private sectors)
	Mental health	(i) “I am currently consulting with a psychologist because I manage a very high level of anxiety and stress.” (I-4, male, 35 years old)
		(j) “The most frequent, precisely that one, [is] that of problems in cultural adaptation, and that is one of those that prevail within these reasons for consultation and that there is a pattern that is repetitive and that one would not only believe that occurs in adults. No, it really occurs in all types of populations.” (KI, psychologist, female, NGO)
		(k) “My house [Venezuela], of course. I was also in my comfort zone, my house, my job, where I lived, what was normal, my routine, my family, my dad [referring to the family in Venezuela]” (I-15, female, 42 years old)
(3) Barriers and facilitators of access to health services	Special Permission for Residence (PEP)	(a) “As we do not have passports, right now, they [the government] are taking out the PEP, unlike those people who have stamped their passports. Since we do not have passports because we cannot get it, we would have to wait for Colombian migration to take a day to get PEP.” (I-8, female, 37 years old)
		(b) “The first barrier is the link. If Venezuelans do not have the temporary or special permit, they cannot link as they want. For me, that is the first barrier. Once they have permission, they can be linked.” (FG-1, dentist, public and private sectors)
	Cost	(c) “No, but here, I have not even been able to go to a dentist due to the same situation that, here, everything is very expensive.” (I-1, female, 45 years old)
	Employment/working conditions	(d) “I was working in Parque Berrío [a main square in the city], in a sewing workshop. They make sheets, cut sheets, make quilts, and all that.” (I-17, male, 31 years old)
		(e) “I work daily selling manillas [a kind of souvenir] on buses. I live with my wife [and] my two daughters. My wife works the same way, selling manillas.” (I-7, male, 33 years old)
		(f) “They [Venezuelans] are afraid of losing their jobs … In order not to miss their job, they don’t want an appointment on a Saturday because they don’t work on Sunday. So, they don’t want to use the sick leave because they don’t want to miss their job because if they miss job, they get fired.” (FG-1, dentist, public and private sector)
	Housing	(g) “Here, to rent a house they don’t allow you with a PEP; you have to have real property, you have to have a guarantor. You have to have all those kinds of documents, and if you don’t have a document to support you, they don’t give you anything. So, it is very difficult to be from one place to another.” (I-17, male, 31 years old)
		(h) “We are in a very little room—how to say, three by three practically … we are eight: she [sister] with her three daughters and I with my family. (I-13, male, 45 years old)
	Affiliation to the General Social Security System (health insurance)	(i) “My mother and grandmother did enter with border permits. They do not have PEP—that is, since they have not taken a census again, I have not been able to get them out. If my mother had the permit, I would already have it affiliated with the EPS [Acronym in Spanish for Empresas Promotoras de Salud, in English means: health maintenance organization]. That is the limitation that I have.” (I-11, male, 40 years old)
		(j) “I am very calm in the aspect of health because I already have my wife and my daughter linked to the EPS—that is, that tranquility that if I need a consultation, I [can] easily ask for it.” (I-11, male, 40 years old)
(4) Support networks	Family	(a) “Well, since he [a friend] has his family here, here, where we have rented, is family. The man above is his cousin.” (EI-1, female, 45 years old)
	Nongovernmental organizations (NGOs)	(b) “The UNHCR [NGO with an office in Medellín] gave me the appointment by phone and told me that I had an appointment here [Famicove, an NGO located in Medellín]. I didn’t even know where it was, even the lady who is seeing herself right now came and was the one who accompanied me to look for the address of Famicove.” (I-E09, female, 56 years old)
	Church	(c) “Those from the church also arrived, how do I say, well, those who work in the rectory also arrived, saw me, [health staff] took my blood pressure. Almost every day, they went to the house, they took my blood pressure, they [health staff] gave me the pill.” (I-6, female, 45 years old)
(5) Expectations	Short term	(a) “Our plans right now … [are] to get something more comfortable for ourselves because since … we are five people who live here and, well, we are very, very small, but we have … plans to go to something bigger.” (I-1, female, 45 years old)
		(b) “Stay as we are right now, look for a good job, why do we do, without a job we can’t do anything else...” (I-17, male, 31 years old)
	Medium term	(c) “My plans for the future [are to] validate my career [and] practice my profession here. I want to do another specialization, but I want to do it here and nothing else. [I want to] get a job in my area.” (I-4, male, 35 years old)
	Long term	(d) “Right now, in the situation we are in [all migrants], we wanted to return, but right now in Venezuela, things are much worse than when we came.” (I-7, male, 33 years old)
		(e) “The truth is, yes, I would like…, but right now, the only way for me to return is not to stay but to go visit, let the government fall, visit the family again, which is what one misses the most.” (I-16, male, 34 years old)

* The alphanumeric code used to identify the source of the verbatim quote is as follows: I—interview, sex, number; KI—key informant, sex, type; FG—focus group, type.

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
