# Peer review of "A Qualitative Study of the Health Perceptions in the Venezuelan Immigrant Population in Medellín (Colombia) and Its Conditioning Factors"

_ijerph, 2021, doi:10.3390/ijerph18083897_

Round 1
Reviewer 1 Report
There seems to be a lack of scientific interest, the result is based on an approximation of the participants’ perceptions of their state of health, except the perception of a defective public health plan for migrants. Further quantitative datas would help to achieve the focus of the manuscript.
Also, you should check the guide for authors for the exact word count of the abstract or the right format to upload images and tables (figure 2 appears pixelated).
Author Response
Please see the attachment
Regards

Reviewer 2 Report
This is an original work.
Two points can be discussed or presented in the M&M paragraph:
- Justify the low numbers of inclusions in the focus groups (n=2)
- Justify the number of strutured interviews (n=17). Was the saturation attained ?
Author Response
Please see the attachment.
Kind regards,

Reviewer 3 Report
An interesting paper. Some corrections/ clarifications suggested in the following lines:
44 should this be “south- south”
217 “Have been identified some aspects”—or “have identified some aspects”
219 Is “symptomatic root tips” referring to root remnants or dental abscess associated with apices of teeth- this is not clear?
220 "these dental emergency care have been"- sounds odd- is it "these dental emergency cases/issues/ problems have been" ... or is it : "Emergency dental care has been the main...."
356 As COVID 19 was mentioned, consideration or some discussion can be given to its impact on interviewee health, presenteeism and its impact on health.
378- 382 this sentence is too long. Can be broken in 2 sentences for easier reading.
Author Response
Please see the attachment
Kind regards,

Reviewer 4 Report
Dear authors,
Thank you for this interesting work and findings. Please pay attention to the following points:
- Line 48 - please translate the Spanish name to English
- Line 50-51 I was not sure if I understood you properly, please considering rephrasing this sentence
- Table 1 has a typo, and also it did not define what is urban and what is rural area. Please consider explaining the boundaries between both groups.
- Figure 2 - please upload a better quality version
- Line 220 - "These" should be "This" I think.
- As long you mentioned the impact of COVID-19 on migrants health and their self-perception of health, you can check these studies they can help enriching your introduction or discussion.
https://doi.org/10.1111/odi.13491
https://doi.org/10.1051/mbcb/2020056
Regards,
Author Response
Please see the attachment
Kind regards,
